# Workshop Submission: Towards Making Untrainable Networks Trainable

**Vighnesh Subramaniam[1]\*, Tomaso Poggio[1], Boris Katz[1], Brian Cheung[1]†, Andrei Barbu[1]†**
[1]MIT CSAIL, CBMM
[1]{vsub851,tp,boris,cheungb,abarbu}@mit.edu

## Abstract

We demonstrate that architectures which traditionally are considered to be ill-suited for a task can be trained using inductive biases from another architecture. Networks are considered untrainable when they overfit, underfit, or converge to poor results even when tuning their hyperparameters. For example, plain fully connected networks overfit on object recognition while deep convolutional networks without residual connections underfit. The traditional answer is to change the architecture to impose some inductive bias, although what that bias is remains unknown. We introduce guidance, where a guide network guides a target network using a neural distance function. The target is optimized to perform well and to match its internal representations, layer-by-layer, to those of the guide; the guide is unchanged. If the guide is trained, this transfers over part of the architectural prior and knowledge of the guide to the target. If the guide is untrained, this transfers over only part of the architectural prior of the guide. In this manner, we can investigate what kinds of priors different architectures place on untrainable networks such as fully connected networks. We demonstrate that this method overcomes the immediate overfitting of fully connected networks on vision tasks and makes plain CNNs competitive to ResNets.

## 1 Introduction

When creating neural networks, we tend as a community to follow recipes that select among a few architectures known to work for particular tasks [39, 9, 17]. Architecture is critical. The gains made on tasks like object recognition are attributable to imposing an inductive bias, i.e., a prior, on the design of new architectures [19, 2]. Convolutional networks unlocked many vision problems [31, 21] and the recent advent of the Transformer [46, 15, 1] did the same for language. Despite this, finding new architectures and overcoming the limitations of existing architectures remains a sort of "dark art". While an architecture imposes some prior, we often do not fully understand what that prior is. One example of this that remains an open discussion is the role of residual connections in making very deep convolutional networks easier to train [24]. If we fully understood the priors that architectures imposed, we could translate between priors and architectures freely – either by specifying the prior we want and then deriving the appropriate architecture or by dispensing with architectures and imposing the prior directly.

Recent theorems [37] state that for each function which is efficiently Turing computable, there exists a deep network that can approximate it well. Furthermore, a graph representing such a function is compositionally sparse; that is, the nodes of the associated Directed Acyclic graph (DAG) represent constituent functions with a small effective dimensionality. A reasonable conjecture is that neural networks with an architecture which is similar to the DAG of the unknown target function are

---

\*Corresponding author.
†Equal senior contribution

Unifying Representations in Neural Models (UniReps) Workshop at NeurIPS 2024.

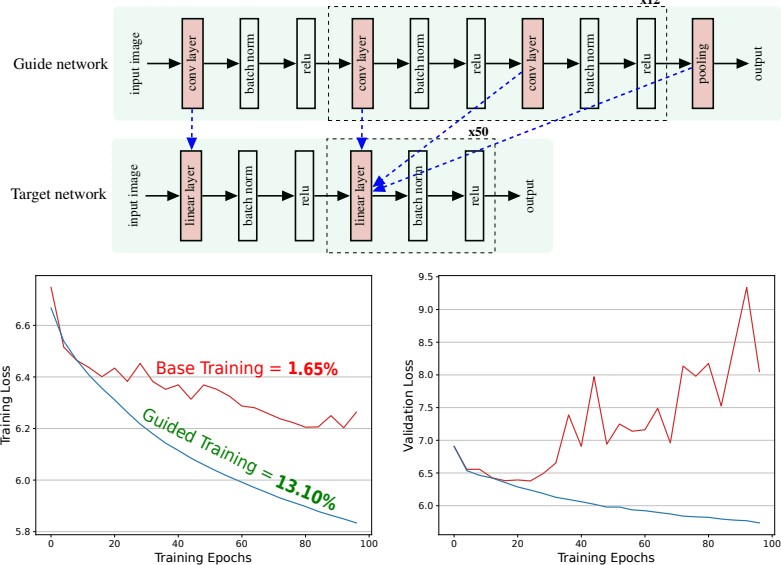

Figure 1: **Guidance between two networks makes untrainable networks trainable**. Given a target which cannot be trained effectively on a task, e.g., a fully connected network which immediately overfits on vision tasks, we guide it with another network. In addition to the target's cross-entropy loss, we encourage the network to minimize the representational similarity between target and guide activations layer by layer. The guide can be untrained, i.e., randomly initialized. This procedure transfers the inductive biases from the architecture of the guide to the target. The guide is never updated. The target undergoing guidance no longer immediately overfits and can now be trained. Here we show an untrained ResNet guiding a deep fully connected network to perform object classification. The FCN alone overfits, the guided version can now be optimized. It has gone from untrainable to trainable.

especially successful in learning it, as it is the case for convolutional networks for image recognition and similar tasks. Because we do not understand the relationship between the kinds of priors on the target functions that different architectures impose, even simple questions have no known answer, such as, does there exist an initialization of an FCN that makes it behave like a CNN, though the graphs of the function they represent are fundamentally different?

To study these problems, we design *guidance*, which is a method to make untrainable networks such as FCNs trainable by transferring the prior of a trainable architecture to the representations of the untrainable architecture. Guidance applies a representational similarity comparison between a target untrainable network and trainable guide network across the activations of several layers of the two networks. We optimize both the task loss of the target network and the representational similarity between the two networks. We find strong improvement of the trainability and performance of our target network on the ImageNet task, indicating an ability to transfer a prior for spatial processing from a convolutional architecture to a fully-connected network.

## 2   Related Work

**Representational Distance**: Our method builds on several metrics that measure distance between high-dimensional activations extracted from neural networks or activity in the brain [28]. Some of these distance metrics make comparisons based on kernel matrices [29, 12, 11] or relative distances [30] between sample representations in a set. Others compute linear [47, 43] or orthogonal projections [3] from one set of representations to another. Others use canonical correlation analysis which finds linear relationships between pairs of vectors [38, 35]. These metrics are designed based on a set of desired invariant properties such as permutation invariance or invariance to linear transformations.

Such approaches have been commonly applied in neuroscience for measuring representational distance of activations from networks and activity in the brain to understand which neural networks are architecturally most similar to the brain [47, 10, 44, 16]. Under this context, [20] has shown the inability of current representational distance metrics to distinguish representations based on architecture. This provides a basis for our experiments to use these metrics to align representations across different neural networks. This paper gives the foundation for our intuition that networks may have similar representations that allow for transferring inductive biases from one network to

another. Recent work has explored the relationship between representational and functional distance of networks, discovering transformations between activations of networks and when these make networks functionally equivalent [28, 6].

**Untrainable Networks**: In this work, we focus on applications of FCNs and plain CNNs for image classification. FCNs have been applied for image classification, where small feed-forward networks were trained on object recognition datasets. These networks were designed with 3-5 hidden layers and less than 100 units per layer [34, 4, 25, 36]. The performance of these networks was very low, where the goal of the paper was to only maximize training fit rather than generalization performance [34, 4]. Other methods were introduced to prevent overfitting in fully connected networks using topological structure [42] or early stopping [7], but these methods had poor fits on the training set due to complex architectural design or complex hyperparameter tuning. Deep convolutional networks have been applied to image classification [31, 5] as well but face problems with vanishing gradients, preventing deep stacking of convolutional networks.

**Model Distillation**: Guidance shares a resemblance with model distillation [22, 18, 41, 23]. Distillation transfers knowledge from a teacher model to a student model by introducing a new component to the loss function that enforces the student model to behave like the teacher model [26, 48]. This can consist of penalizing the KL-divergence between the logit predictions of the student and teacher model.

Representation-based distillation [45, 8, 32] and alignment techniques have been proposed to improve alignment between two networks. Certain works have proposed contrastive loss functions on output representations to distill teacher information into the representation space of the student. Other methods introduce correlation congruence or similarity preserving metrics for aligning two networks. Methods have been proposed that use CKA as an alignment approach between representations of two networks or with representations in the brain with notable improvement in network performance [40, 13].

We distinguish guidance from distillation. Guidance can use a smaller untrained guide instead of a larger trained teacher. This is due to guidance operating over intermediate activations of the network rather than the output of the network probabilities or output features, like distillation does. Guidance also operates at many levels at the same time, aligning many layers at once. This helps address the credit assignment problem that gradient descent has when tuning weights early in a network. We also consider many more networks for guidance than is traditional for distillation including networks which have very different architectures like Transformers to RNNs. Distillation is usually carried out between two closely related architectures. We apply guidance to do the opposite.

## 3 Methods

Guidance introduces a term in the loss of a target network, $\mathcal{N}^T$, to encourage representational alignment with a guide network, $\mathcal{N}^G$. Only the parameters, $\theta^T$, of the target are updated — the guide's parameters, $\theta^G$, remain fixed. Per minibatch, representational similarity is computed between the activations of $i^G$th layer of the guide, $\mathbf{A}_{i^G}^G(\theta^G)$, and activations from a corresponding layer $i^T$ of the target, $\mathbf{A}_{i^T}^T(\theta^T)$. We refer to the correspondence between layers of the guide $\{i^G\}$ and layers of the target $\{i^T\}$ as $I$. While this correspondence, $I$, could be complex as any two architectures can form a guide/target pair, here we choose architectures that make the correspondence obvious as is discussed later. For example, the stacked RNNs and Transformers have the same number of layers in our experiments.

The target and guide receive the same input. Per minibatch, we collect activations from intermediate layers of both networks. Layers of guide network are mapped to layers of the target network. We formulate the loss in terms of minimizing the *representational dissimilarity*, $\bar{\mathcal{M}}$, i.e., the complement of a representational similarity metric, between guide and target activations layer by layer, summing the results. We only consider the centered kernel alignment, CKA, between the activations in this publication. Many other possibilities exist. Any representational similarity function which is differentiable could be used. Efficiency or incremental computation is much more important than it is in traditional applications since this operation happens for every minibatch.

Given $\mathcal{L}_T$ as the original loss of the target network, the guide network's original loss function is irrelevant. The guide need not even have been trained on the same task or the same dataset. It need not even have been trained at all. This latter setting is what allows transferring architectural priors without transferring knowledge from the guide to the target, as there is none in a randomly initialized

guide. The final loss we optimize, $\mathcal{L}$ is:

$$\mathcal{L}(\theta^T) = \mathcal{L}_T(\theta^T) + \sum_{i \in I} \bar{\mathcal{M}}(\boldsymbol{A}_{i^T}^T(\theta^T), \boldsymbol{A}_{i^G}^G(\theta^G)) \tag{1}$$

This minimizes a task loss while increasing alignment between the target and guide networks given the mapping between them. The mapping may be sparse, not every layer needs to be used. This is important for guidance with transformers or stacked RNNs as will be explained later. We don't incorporate any weight on the layer-wise similarity component of the loss. Note that the guide's parameters, $\theta^G$, are constants, i.e., the guide is never updated.

## 4 Experiments

We design several settings with different target and guide networks to thoroughly test our approach. We include image-based settings and sequence modeling-based settings as well. In choosing target networks, we consider a broad range of designs for networks that are not traditionally applied (e.g., an MLP in image classification). We consider these networks as "untrainable" for a wide range of reasons such as difficulties with getting a good fit on the training set, overfitting on the validation set, and/or poor test accuracy regardless of fit on both sets. In most settings, these issues are driven by algorithmic constraints of the target network, which becomes the inspiration for our design.

We focus on image classification and use the ImageNet-1K [14] dataset for training and testing. We use the splits defined by the dataset. We report accuracy on the validation set for all experiments.

We use three target networks: Deep FCN, Wide FCN, and Deep ConvNet. Deep FCN is a fully-connected network with 50 blocks consisting of feedforward layers followed by non-linearities. This network is an untrainable architecture, lacking inductive biases to prevent overfitting and having vanishing gradients. Wide FCN is a fully connected network with 3 blocks with feedforward layers that have 8192 units. This is categorized as an untrainable task due to a saturation in the training performance. Deep ConvNet is the same architecture as ResNet-50 [21], but without residual connections. This is categorized as an untrainable architecture due to the vanishing gradient problem. We use two guide networks: ResNet-18 and ResNet-50. ResNet-18/50 is a deep convolutional network with 18/50 convolutional blocks and residual connections. We refer to **(author?)** [21] for ResNet 18/50 design. We supervise the Deep FCN and Shallow FCN with ResNet-18 and supervise the Deep ConvNet with ResNet-50.

For each setting, we train the base target network and an experiment where a guide network supervises the base target network. All networks are trained with cross-entropy loss. We use the Adam [27] optimizer with a learning rate of 1e-4.

## 5 Results

We show image classification results with our three networks in Figure 2 and Table 1.

Across all our networks, we observe significant improvement from using a guide network to provide representational guidance. We see significant accuracy gains of 5-10% on the test performance. We also observe significantly better loss curves from a better fit with the training loss and less overfitting with the validation loss. Most interestingly, we highlight that using a randomly initialized guide network can perform better than using a trained guide network. For example, the Deep FCN results in Figure 2 is significantly better with a randomly initialized ResNet-18 as the guide network instead of a trained ResNet-18. This trend also occurs with Wide FCN.

We note that this trend is not entirely consistent as indicated by the Deep ConvNet. One potentially explanation is that the architectural prior is the same for both the Deep ConvNet and ResNet-50. This explanation provides an additional interpretation for the role of residual connections and their influence on the representation space. Specifically, the power of residual connections are not just architectural, but training-based.

Overall, these experiments establish that these three networks which are usually not applicable to image classification can be operationalized under our training approach, especially with randomly initialized networks.

| Experiment | ImageNet Top-5 Validation Accuracy ($\uparrow$) |
|---|:---:|
| ResNet-18 | 89.24 |
| Untrained ResNet-18 | $0.24 \pm 0.043$ |
| ResNet-50 | 92.99 |
| Untrained ResNet-50 | $0.54 \pm 0.029$ |
| Deep FCN | $1.65 \pm 0.51$ |
| ResNet-18 $\rightarrow$ Deep FCN | $7.50 \pm 1.51$ |
| Untrained ResNet-18 $\rightarrow$ Deep FCN | $\mathbf{13.10} \pm 0.72$ |
| Wide FCN | $34.09 \pm 1.21$ |
| ResNet-18 $\rightarrow$ Wide FCN | $\mathbf{43.01} \pm 0.92$ |
| Untrained ResNet-18 $\rightarrow$ Wide FCN | $39.47 \pm 0.31$ |
| Deep ConvNet | $70.02 \pm 1.52$ |
| ResNet-50 $\rightarrow$ Deep ConvNet | $\mathbf{78.91} \pm 2.16$ |
| Untrained ResNet-50 $\rightarrow$ Deep ConvNet | $68.17 \pm 2.54$ |

Table 1: **Guidance improves performance for image classification**. Alignment with a ResNet dramatically improves a deep FCN, particularly with an untrained ResNet. Significant gains are seen with a wide FCN as well. Deep CNNs without residuals gain only with a trained ResNet. Across all settings, guidance can help train architectures that were otherwise considered unsuitable.

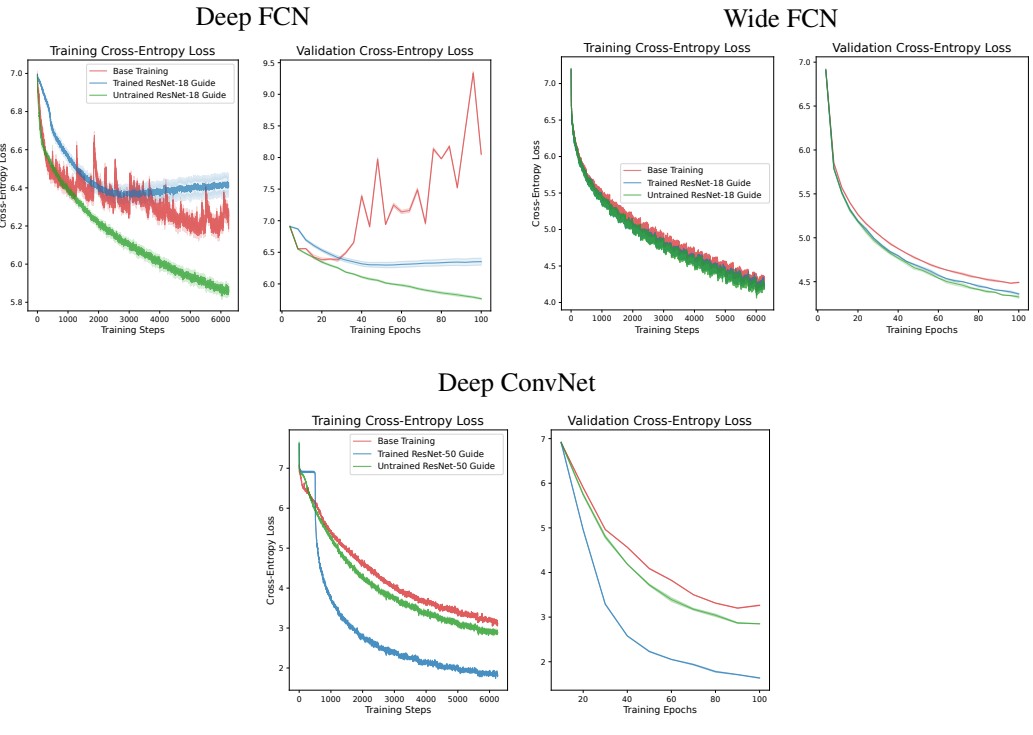

Figure 2: **Untrainable Networks for Object Detection**. We train three networks on ImageNet both without access to a guide network and with access to a trained and untrained guide network. Base training results are shown in red, our approach with a trained guide network is shown in blue and our approach with an untrained guide network is shown in green.

## 6 Conclusion

We introduced guidance, a new method for neural network training to make untrainable networks trainable. We applied our method to fully-connected networks and deep convolutional networks trained on ImageNet, guided by ResNet-18 and ResNet-50. Our method is simple and scalable, applicable to any pair of guide and target networks across many tasks. This method allows for studying unknown properties of neural networks related to initialization or optimization and their relation to the prior inductive biases of a network. Characterizing architectural priors and untrainable networks can be made more precise using our methodology.

**Acknowledgments**

This work was supported by the Center for Brains, Minds, and Machines, NSF STC award CCF-1231216, the NSF award 2124052, the MIT CSAIL Machine Learning Applications Initiative, the MIT-IBM Watson AI Lab, the CBMM-Siemens Graduate Fellowship, the DARPA Artificial Social Intelligence for Successful Teams (ASIST) program, the DARPA Mathematics for the DIscovery of ALgorithms and Architectures (DIAL) program, the DARPA Knowledge Management at Scale and Speed (KMASS) program, the DARPA Machine Common Sense (MCS) program, the United States Air Force Research Laboratory and the Department of the Air Force Artificial Intelligence Accelerator under Cooperative Agreement Number FA8750-19-2-1000, the Air Force Office of Scientific Research (AFOSR) under award number FA9550-21-1-0399, and the Office of Naval Research under award number N00014-20-1-2589 and award number N00014-20-1-2643. The views and conclusions contained in this document are those of the authors and should not be interpreted as representing the official policies, either expressed or implied, of the Department of the Air Force or the U.S. Government. The U.S. Government is authorized to reproduce and distribute reprints for Government purposes notwithstanding any copyright notation herein.

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

# A   Guide-Target Network Mapping Methods

## A.1   Centered Kernel Alignment

To compare representations, we use a representation similarity metric, $\mathcal{M}$, which corresponds to centered kernel alignment (CKA) [29] in our setting. We specifically consider linear CKA.

CKA uses kernel functions on mean-centered representations to compute representational similarity matrices, which are then compared via the Hilbert-Schmidt Independence Criterion (HSIC). More specifically, suppose we have two sets of representations $\boldsymbol{R} \in \mathbb{R}^{b \times d_1}$ and $\boldsymbol{R}' \in \mathbb{R}^{b \times d_2}$. We first compute the Gram matrices for each set of representations

$$\boldsymbol{K} = \boldsymbol{R}\boldsymbol{R}^T, \boldsymbol{L} = \boldsymbol{R}'\boldsymbol{R}'^T \tag{2}$$

We center the Gram matrices by introducing a matrix, $H$ where $H = \boldsymbol{I}_b - \frac{1}{n}\boldsymbol{1}\boldsymbol{1}^T$.

$$\tilde{\boldsymbol{K}} = \boldsymbol{H}\boldsymbol{K}\boldsymbol{H}, \tilde{\boldsymbol{L}} = \boldsymbol{H}\boldsymbol{L}\boldsymbol{H} \tag{3}$$

We compute the HSIC on the Gram matrices.

$$HSIC(\boldsymbol{K}, \boldsymbol{L}) = \text{tr}(\tilde{\boldsymbol{K}}, \tilde{\boldsymbol{L}}) \tag{4}$$

Finally, we define our linear CKA metric as:

$$\mathcal{M}(\boldsymbol{R}, \boldsymbol{R}') := \text{CKA}(\boldsymbol{K}, \boldsymbol{L}) = \frac{HSIC(\boldsymbol{K}, \boldsymbol{L})}{\sqrt{HSIC(\boldsymbol{K}, \boldsymbol{K}) * HSIC(\boldsymbol{L}, \boldsymbol{L})}} \tag{5}$$

In our setting, we consider representational *dissimilarity* and aim to minimize the dissimilarity between representations from our target network and guide network. We define this as:

$$\bar{\mathcal{M}}(\boldsymbol{R}, \boldsymbol{R}') = 1 - \mathcal{M}(\boldsymbol{R}, \boldsymbol{R}') \tag{6}$$

Linear CKA ranges from 0 (identical representations) to 1 (very different representations). Because of this, we take the complement by subtracting the linear CKA from 1 to represent dissimilarity.

## A.2   Layerwise Mapping

We design a simple method for mapping guide layers to target layers as part of providing supervision. The goal of this method is to make guide and target networks architecturally agnostic i.e. we can supervise any target network with any guide network.

As a simple approach, we evenly spread layer computations of our guide network over our target network. For example, if we consider ResNet-18 and a 50-layer FCN, we would map every ResNet layer to every second or third linear layer of the MLP. The intuition for this approach follows from the aim of discovering the same function of the guide network using a target network. Through the design of evenly spreading layers of our ResNet-18, we are guiding the MLP to find learn a function similar to guide network.

For our mapping, we consider activations from layers with tunable weights i.e. convolutional, linear, or LSTM/RNN based layers. For multiple stacked RNNs, LSTMs, or transformers, we extract feature representations from intermediate layers in the stack as well. Skipping layers based on non-linear transformations reduces memory overhead associated with storing representations per batch.

## A.3 Algorithmic Description

We provide an algorithmic description of guidance in Algorithm 1.

---

**Algorithm 1 Guidance**: Guide Network Representational Alignment

---

**Require:** Target network; $\mathcal{N}^T$ with parameters $\theta^T$; Guide network $\mathcal{N}^G$; Dataset $\mathcal{D} = \{(x_i, y_i)\}_{j=1}^N$; Representational Distance Metric $\bar{\mathcal{M}}$; Loss function $\mathcal{L}^T$

1:  **for** $j = 1 \rightarrow N$ **do**
2:      # Base training with vanilla loss function
3:      outputs $\leftarrow \mathcal{N}^T(x_j)$
4:      loss $\leftarrow \mathcal{L}_T(\text{outputs}, y_j \mid \theta^T)$
5:      # collect layer activations
6:      $\{A_{iT}^T\}_{iT=1}^t \leftarrow \text{activations}(\mathcal{N}^T(x_j))$
7:      $\{A_{iG}^G\}_{iG=1}^l \leftarrow \text{activations}(\mathcal{N}^G(x_j))$
8:      # Get step size between the number of layers between the two networks for layer mapping.
9:      **if** $l > 1$ **then**
10:          $step \leftarrow (t-1)/(l-1)$
11:      **else**
12:          $step \leftarrow 1$
13:      **end if**
14:      # Map the layers and add up layer-wise representational distance
15:      total $\leftarrow 0$
16:      **for** $i = 1 \rightarrow l$ **do**
17:          index $\leftarrow \min(\text{round}(i \times step), l-1)$
18:          rep $\leftarrow \mathcal{M}(A_{\text{index}}^T, A_{\text{index}}^G)$
19:          total $\leftarrow$ total $+$ rep
20:      **end for**
21:      loss $\leftarrow$ loss $+$ total
22: **end for**

---

# B   Methodology Limitations

Our work has a number of limitations. We aimed for initial improvements of trainability instead of maximal performance on any one task. This would have required us to carefully tune the hyperparameters involved. We preferred to show how guidance works in general rather than in cherry-picked or carefully tuned settings. To that end, we also did not optimize networks to convergence, nor did we attempt to experiment with other optimizers. Once we reproduced a well-known problematic training phenomenon, we showed that it could be overcome. We consider a network trainable enough to overcome a problem when the original problem disappears. For example, successfully training fully connected networks for object recognition was hopeless because they immediately overfit; using our guidance method they no longer do so. This does not mean that they are necessarily useful as object recognizers at present. In the case of fully connected networks, their present performance with guidance training is too low, but with additional work, we believe their performance could be substantially increased now that their train and test loss are moving in the right direction. In some cases, by applying guidance, we do see large useful improvements.

Our guide network supervision through representational alignment has one primary limitation due to increased memory usage during training. Due to saving activations across several layers of the two networks, GPU memory usage increases dramatically. Moreoever, our methodology works better as batch size increases since this allows for better approximation of representational similarity, increasing memory usage even more. Furthermore, including more layers for supervision leads to improved results.

In this paper, we introduce simple techniques to handle memory constraints such as gradient accumulation and gradient checkpointing [33]. In practice, more memory optimization techniques may become necessary to consider larger untrainable networks. Further work could consider using

stronger representational alignment strategies to reduce the number of samples necessary to achieve a strong fit.

