# OpenReview forum: "Workshop Submission: Towards Making Untrainable Networks Trainable"
_NeurIPS.cc/2024/Workshop/UniReps — UniReps_

### Official Review · Reviewer_WqZa · 2024-10-03
**Interesting narrative, but need proper references for better context**

**Rating:** 6
**Confidence:** 4

**Review:**

The paper introduces a method to make "untrainable" neural networks trainable by using a source network as a guide. Essentially, this closely aligns with the traditional teacher-student framework, where the source (teacher) network supervises the target (student) network through representational alignment.

While the paper presents an interesting perspective, particularly in applying these concepts to traditionally difficult-to-train architectures, its novelty is more in its framing than in the core approach.

The paper makes some notable contributions, including the interesting finding that randomly initialized source networks sometimes outperform trained ones - a phenomenon that distinguishes this work from typical teacher-student approaches. However, the paper does not adequately address the connection between its method and the well-established teacher-student framework, which is a significant oversight.

The related work in the introduction is notably lacking in terms of references to existing work on knowledge distillation, model transfer, and other teacher-student approaches, which would have provided clearer context and differentiation. Furthermore, while the experiments demonstrate meaningful improvements across several different types of traditionally difficult-to-train architectures, they are not sufficiently benchmarked against existing teacher-student methods, making it difficult to assess the true contribution relative to the SOTA. Without these comparisons, the claim of significant performance improvements remains isolated and lacks grounding in the broader literature.

Suggestions to the authors to improve their paper:
- Add related work references to address the connections with existing teacher-student frameworks.
- Include comparisons to SOTA methods in the teacher-student paradigm.

In conclusion, while the work presents an interesting narrative and shows promising results for improving traditionally difficult architectures, it needs stronger positioning and benchmarking against relevant, existing frameworks to better highlight its contributions.

---

### Official Review · Reviewer_y18p · 2024-10-04
**The authors propose an effective method to train typically hard-to-train networks (targets) by aligning their features with those of a more suitable architecture (source). While the approach is novel, its practical utility is unclear. The method requires training the target while utilizing the intermediate features from the source network. This raises questions about the end-product's value compared to simply using the source network.**

**Rating:** 5
**Confidence:** 2

**Review:**

Clarification needed on training with untrained source network setting:

1. Are both source and target networks trained in parallel, with an additional alignment loss for the target?
2. Or are features from a randomly initialized source network used to align the target's features?

If it's the latter, effectiveness seems unlikely as randomly initialized CNN features, while structured, would be unrelated to the downstream task. Please clarify the exact procedure and its rationale.

**Motivation concerns:**
The paper's motivation for training disadvantaged architectures (e.g., MLPs) is unclear. The method uses a pretrained source network (e.g., CNN) on the same dataset to guide the target network. This raises two key issues:

- The source network likely outperforms the target network.
- It's unclear why one would use the target network over the already trained source network.

**Experimental comments**

- Aligning the intermediate features using alternative similarity metrics (e.g. cosine similarity, mse) could be interesting.
- Effect of aligning only the penultimate layer features instead can be used as a lightweight comparison.

---

### Official Review · Reviewer_6ex4 · 2024-10-06
**Developed an interesting approach to improve the performance of otherwise underperforming neural networks. These underperforming neural networks are Deep FCN, Shallow FCN and Deep ConvNet.**

**Rating:** 6
**Confidence:** 1

**Review:**

- Authors have taken inspiration from neuroscience to develop the approach of making untrainable networks trainable. They have defined a method that introduces a source network (one that acts as a guide) and a target network (one that is untrainable). They have optimized both task loss and representational similarity between the source and target network. The task is clearly defined. The results of the experiments suggest an improvement. The approach provides a novel way of understanding these untrainable networks.
- The task has been performed on a limited number of target networks. It would have been interesting to see how one can scale this approach to various other untrainable networks.

---

### Decision · Program_Chairs · 2024-10-10

**Decision:**

Accept

**Comment:**

In light of the positive reviewers' feedback and relevancy of the submission, we are pleased to accept this paper for presentation at UniReps 2024. We kindly ask the authors to incorporate the reviewers' suggestions and feedback in the final camera-ready version of the manuscript.